# Association between SGLT2 Inhibitors and Cardiac Rehabilitation Outcomes in Patients with Cardiovascular Disease and Type 2 Diabetes Mellitus

**DOI:** 10.3390/jcm11195956

**Published:** 2022-10-09

**Authors:** Ayuko Kashima, Kentaro Kamiya, Nobuaki Hamazaki, Kensuke Ueno, Kohei Nozaki, Takafumi Ichikawa, Masashi Yamashita, Shota Uchida, Takumi Noda, Kazuki Hotta, Emi Maekawa, Minako Yamaoka-Tojo, Atsuhiko Matsunaga, Junya Ako

**Affiliations:** 1Department of Rehabilitation Sciences, Graduate School of Medical Sciences, Kitasato University, Sagamihara 252-0373, Japan; 2Department of Rehabilitation, School of Allied Health Sciences, Kitasato University, Sagamihara 252-0373, Japan; 3Department of Rehabilitation, Kitasato University Hospital, Sagamihara 252-0375, Japan; 4Division of Research, ARCE Inc., Sagamihara 252-0306, Japan; 5Exercise Physiology and Cardiovascular Health Laboratory, Division of Cardiac Prevention and Rehabilitation, University of Ottawa Heart Institute, Ottawa, ON K1Y 4W7, Canada; 6Department of Cardiovascular Medicine, School of Medicine, Kitasato University, Sagamihara 252-0373, Japan

**Keywords:** sodium-glucose cotransporter 2 (SGLT2) inhibitor, cardiovascular disease (CVD), type 2 diabetes mellitus (T2DM), cardiac rehabilitation (CR), physical function

## Abstract

The use of sodium-glucose cotransporter 2 (SGLT2) inhibitors in patients with type 2 diabetes mellitus (T2DM) has been associated with decreased skeletal muscle mass but remains unclear in patients with cardiovascular disease (CVD) undergoing comprehensive outpatient cardiac rehabilitation (CR). Therefore, this study investigates the effect of SGLT2 inhibitors on the outcomes of patients with CVD and T2DM undergoing comprehensive outpatient CR. The study included 402 patients with CVD and T2DM who participated in comprehensive outpatient CR. Physical functions (grip strength, maximal quadriceps isometric strength, usual gait speed, and 6-minute walking distance) were measured at discharge as baseline and 5 months thereafter, and the association between physical functions and SGLT2 inhibitor use was reviewed. Physical functions improved regardless of SGLT2 inhibitor use. Multiple regression analysis showed that SGLT2 inhibitor use was not associated with improvement or decline in physical functions (*p* ≥ 0.05). The use of SGLT2 inhibitors in patients with CVD and T2DM undergoing outpatient CR did not impair improvement in physical functions.

## 1. Introduction

Exercise-based outpatient cardiac rehabilitation (CR), including exercise therapy, is an established non-pharmacological therapy that has shown improved outcomes in patients with cardiovascular disease (CVD) and heart failure (HF) in previous studies [1]. CR guidelines recommend it as equivalent (Class 1, Level A) to pharmacotherapy [2,3,4]. Recently, sodium-glucose cotransporter 2 (SGLT2) inhibitors, insulin-independent hypoglycemic agents developed for the treatment of diabetes, have attracted much attention [5]. The effects of SGLT2 inhibitors have been reported in patients with CVD [6,7] and SGLT2 inhibitors have been added to class I standard medical therapy for patients with HF [2,8].

However, previous studies on type 2 diabetes mellitus (T2DM) have shown a decrease in skeletal muscle mass with SGLT2 inhibitors [9,10,11]. In addition to reduced skeletal muscle mass, mice with adjusted dietary intake have also shown muscle weakness [12]. However, it remains unclear as to how SGLT2 inhibitors affect changes in skeletal muscle mass and physical function, while there is concern that SGLT2 inhibitors may reduce physical function. Although several patients with CVD undergoing CR are increasingly prescribed SGLT2 inhibitors, the impact of SGLT2 inhibitor administration on the physical outcomes of outpatient CR remains nebulous.

Therefore, this study determines the association between SGLT2 inhibitor use and changes in physical function in patients with CVD and T2DM who participated in outpatient CR.

## 2. Materials and Methods

In this single-center retrospective observational study, a cohort of consecutive patients with CVD admitted to Kitasato University Hospital for CVD treatment from April 2007 to August 2020 were reviewed. Among them, those who also had T2DM and for whom at least one physical function (grip strength, maximal quadriceps isometric strength [QIS], usual gait speed, or 6-minute walking distance [6 MWD]) could be measured longitudinally twice were studied. Patients who were taking SGLT2 inhibitors prior to admission were excluded. This research protocol was conducted in accordance with the ethical guidelines of the Declaration of Helsinki and was approved by the Kitasato Institute Clinical Research Review Committee (KMEO B18-075).

### 2.1. Patient Characteristics

We obtained data on age, sex, body weight, height, body mass index (BMI), left ventricular ejection fraction (LVEF), number of outpatient rehabilitation participations, HF severity according to the New York Heart Association functional classification (NYHA class) and history of HF and acute myocardial infarction from medical records at admission. Hemoglobin (Hb), B-type natriuretic peptide, and serum creatinine (sCr) levels were analyzed as routine tests. The estimated glomerular filtration rate (eGFR) was determined from the sCr value using 194 × sCr − 1.094 × age (years) − 0.287 × 0.739 (for women) developed by the Japanese Society of Nephrology [13]. Further, 2-dimensional method was used to measure LVEF on echocardiogram.

### 2.2. SGLT2 Inhibitors

The patients in our study using SGLT2 inhibitors were prescribed any of the following four drugs: empagliflozin, dapagliflozin, canagliflozin, and luseogliflozin. Dosage and drug withdrawal were performed according to the relevant documentation.

### 2.3. Physical Function

Physical functions including grip strength, QIS, usual gait speed, and 6 MWD were assessed at discharge as baseline, at the end of CR, and 5 months after the start of CR.

Grip strength was measured using a grip strength meter (TKK 5101 Grip-D; Takei, Tokyo, Japan) to evaluate the upper extremity muscle strength. In the grip strength measurement, the grip width was adjusted so that the proximal interphalangeal joint of the index finger was at 90°, the patient was placed in a sitting position with the elbow joint angle flexed to 90°, and the grip strength meter was held outward. A 3 s strong grip was performed twice, alternately on the left and right sides. The examiner lightly supported the patient’s elbow and the tip of the grip strength tester alternately on the left and right sides and instructed the patient not to hold their breath during the measurement to avoid the Valsalva maneuver. The highest values of the left and right grip strength were expressed as the mean value (kg) [14].

To assess lower limb muscle strength, QIS was measured using a hand-held dynamometer (μ-Tas; ANIMA, Tokyo, Japan). For the QIS measurements, the patient was seated with the hip and knee joints in 90° flexion and the thigh was fixed to a chair with a band. A dynamometer was placed on the anterior surface of the lower leg and above the two transverse fingers of the external tibia and maximal isometric contractions of the quadriceps were performed twice for 5 s, on alternating left and right sides. To avoid the Valsalva maneuver, patients were instructed not to hold their breath during contraction and to avoid compensatory movements as much as possible. The highest value of strength on each side was expressed as an average (kgf) [15].

For usual gait speed, patients were asked to walk 16 m at usual gait speed, then walk from 3 to 13 m at 10 m intervals, and the time required was measured excluding the 3 m acceleration and deceleration times before and after the walk. A stopwatch was used for the measurements and the timing for starting and stopping the stopwatch was when the subject stepped on or over the 3 and 13 m lines, respectively. Measurements were taken twice and the faster walking time was adopted to calculate the gait speed (m/s) [16].

The functional capacity was assessed using the 6 MWD. The walking distance was calculated with the help of a stopwatch by counting the number of times the subject walked back and forth along a 20 m hallway in 6 min. The patients were instructed to walk as much distance as possible in 6 min and could take as many breaks as needed during the test [17].

### 2.4. Endpoints

The primary endpoint was the amount of change from baseline to 5 months (Δ physical function) in physical function (grip strength, QIS, usual gait speed, and 6 MWD). The secondary endpoint was the change in body weight (Δ body weight), calculated as difference between physical function at 5 months after baseline and baseline physical function for Δ physical function and difference between body weight at 5 months after baseline and baseline body weight for Δ body weight.

### 2.5. Cardiac Rehabilitation Program

Comprehensive outpatient CR was implemented according to the statement of the Japanese Cardiovascular Society and education on self-management of medication, nutrition, and exercise was provided by a cardiologist and medical staff prior to discharge [3]. Participation in outpatient CR at least once a week and self-exercise 3–5 times a week were recommended. Exercise training in the outpatient CR consisted of a 5 min warm-up, 20–40 min of aerobic exercise at an intensity of 12–14 on the Borg scale on a treadmill or bicycle ergometer, and a 3 min cool-down [18]. All in-hospital exercise therapies were supervised by a trained nurse or a physical therapist and included heart rate and electrocardiographic monitoring with a monitored electrocardiogram. In addition, blood pressure and Borg scale assessments were performed every 5–10 min. The total number of outpatient CR attended by each participant was collected from the medical records and analyzed.

### 2.6. Statistical Analysis

Statistical analyses were performed using Stata software (version 17, StataCorp LLC, College Station, TX, USA) and EZR version 1.37 (Saitama Medical Center, Jichi Medical University, Saitama, Japan), which is a graphical user interface for R (R Foundation for Statistical Computing, Vienna, Austria). Categorical and continuous data of the baseline characteristics were presented as median (interquartile range), *n* (%). Fisher’s exact test was used to compare categorical variables between the patients using and not using SGLT2 inhibitors. Mann–Whitney *U* test was used to compare continuous variables between the two groups. We used a mixed-effect model and multiple regression model to identify the relationship between SGLT2 inhibitor use and the amount of change in physical function (Δ grip strength, Δ QIS, Δ usual gait speed, and Δ 6 MWD). In addition, we used a linear mixed-effect model and a multiple regression model to identify the relationship between SGLT2 inhibitor use and the amount of change in body weight. In the multiple regression analyses, we used SGLT2 inhibitors, age, sex (male), BMI, NYHA class ≥ III, LVEF, HF, Hb, year of hospitalization (≥2014 or not), and baseline of each physical function as a covariate. In Japan, SGLT2 inhibitors were approved for the treatment of T2DM in 2014. Considering this, we included whether the year of admission was 2014 or later as a covariate. A *p*-value of <0.05 was considered statistically significant [19].

## 3. Results

Patients with CVD and T2DM who participated in comprehensive outpatient CR and were taking SGLT2 inhibitors prior to admission (*n* = 14) were excluded from the study. As a result, 402 patients were included in the analysis (Figure 1). The median age of patients in this study was 69 years, 274 (68.2%) were males, and 32 (8.0%) were SGLT2 inhibitor users (Table 1).

### 3.1. Association between SGLT2 Inhibitor Use and Change in Physical Function

In the mixed-effects model, a significant main effect of time showed an increase in physical functions (all *p* < 0.001), and no significant difference in change in physical function with and without SGLT2 inhibitor use was observed (time × SGLT2 inhibitor; all *p* ≥ 0.05) (Figure 2). Multiple regression analysis showed that SGLT2 inhibitor use was not significantly associated with the change in physical functions (Table 2). 

### 3.2. Association between SGLT2 Inhibitor Use and Change in Body Weight

In the mixed-effects model, no significant main effect of time showed an increase in body weight (*p* = 0.188), and no significant difference in change in body weight with and without SGLT2 inhibitor use was observed (time × SGLT2 inhibitor; *p* = 0.493) (Appendix A). Multiple regression analysis showed that SGLT2 inhibitor use was not significantly associated with change in body weight (Appendix A).

## 4. Discussion

The primary findings of our study were as follows: (1) improvement in physical function was observed in CVD and T2DM patients who participated in an exercise-based CR with or without SGLT2 inhibitor use and (2) there was no significant difference in the amount of change in physical function between the patients using and not using SGLT2 inhibitors. These findings suggest that the favorable effect of exercise-based CR on physical function was equally manifested in patients with CVD and T2DM, with or without SGLT2 inhibitor use. Our study is consistent with a previous study, which reported that exercise-based CR contributed to the recovery of physical function in HF patients, with or without T2DM [20]. Previous studies examining the effect of SGLT2 inhibitor use on exercise tolerance in nondiabetic patients with HF have shown that the prescription of SGLT2 inhibitors contributed to the improvement in exercise tolerance (peak VO_2_, 6 MWD) [21,22]. However, it remains unclear whether prescribing SGLT2 inhibitors affected the functional recovery associated with CR. To the best of our knowledge, this is the first study to show that the use of SGLT2 inhibitors in patients with CVD and T2DM does not interfere with the improvement in outcomes regarding skeletal muscle function in outpatient CR.

SGLT2 inhibitors are effective for improving the prognosis of patients with HF [23,24]. However, there is concern regarding skeletal muscle loss caused by SGLT2 inhibitors [25]. SGLT2 inhibitors significantly reduced body weight and approximately two-thirds of the weight loss was from fat mass and a third from lean body mass [9,26]. Non-randomized observational studies found significant reductions in lean body mass and fat mass in Japanese patients with T2DM after the initiation of SGLT2 inhibitors [10,11]. Other previous studies have reported no effect of SGLT2 inhibitors on lean mass [25,26,27,28]. A marked decrease in lean mass was observed with high doses of canagliflozin [9]. Several researchers have investigated the effect of SGLT2 inhibitors on skeletal muscle mass and function in mice [12,27,28]. Interestingly, Bamba et al. showed that the db/db young mice fed a diet with SGLT2 inhibitors had higher skeletal muscle mass [28], while Nambu et al. found no significant effect in muscle mass of HF model mice [27]. Contrary to these studies, Otsuka et al. reported that when the amount of food in SGLT2 inhibitor-treated mice was adjusted to that in vehicle-treated mice, muscle mass and function were reduced in the treated mice [12]. Muscle mass reduction was found in the tibialis anterior and extensor digitorum longus muscles but not in the soleus muscle, indicating a muscle-fiber-type-specific change [12]. When food intake in the treated group was intentionally adjusted to the control group, protein degradation exceeded protein synthesis as a catabolic response to SGLT2 inhibition, likely causing skeletal muscle mass loss in mice [12]. These studies have raised concerns that SGLT2 inhibitors may induce muscle atrophy. Neither skeletal muscle mass nor lean body mass was measured in our study; however, we confirmed no difference in the amount of change in body weight between SGLT2 inhibitor users and non-users (Appendix A). Body weight is the simplest indicator of nutritional status and significant weight loss is a criterion for frailty [29]. In particular, unintentional weight loss of more than 10 pounds in the previous year or more than 5% of the previous year’s weight at follow-up is suspected to be due to frailty. Considering these previous studies and the present study, it is unlikely that SGLT2 inhibitors induced significant loss of skeletal muscle during 5 months of CR.

The effect of SGLT2 inhibitor use on physical functions, such as muscle strength, gait speed, and walking capacity, is unclear. Deterioration in physical function and muscle atrophy, as well as the subsequent inhibition of recovery, has been concerns in terms of potential side effects from SGLT2 inhibitors. Our results partially addressed these concerns. In this study, we showed that the physical function of SGLT2 inhibitor users was ameliorated after 5 months of exercise-based CR and the degree of recovery was not different from that of non-prescribers. These findings suggest that SGLT2 inhibitors do not interfere with exercise-based CR-induced recovery of physical function. In addition to body weight, grip strength [30,31], QIS [32,33], gait speed [34,35], and 6 MWD [36] are prognostic factors. Our study found as much improvement in physical function as in the SGLT2 inhibitor non-user group, even if SGLT2 inhibitor use. Taken together with the fact that body weight was not altered, our results suggest that SGLT2 inhibitors had no adverse effects on skeletal muscle. We also found significant improvements in gait speed and 6 MWD. Gait speed and 6 MWD are useful for identifying difficulties in activities of daily living (ADL) in patients with CVD [37]. Therefore, SGLT2 inhibitors may not inhibit improvement in ADL. Linden et al. investigated the effects of co-treatment with an SGLT2 inhibitor and exercise training on exercise capacity of T2DM model rats [38]. The authors showed that rats that were administered the SGLT2 inhibitor and underwent exercise training had better exercise capacity than those that underwent exercise alone [38]. However, in this study, we found no additional value of SGLT2 inhibitors for improving exercise tolerance with CR. Differences in results between Linden’s study and our study may be attributed to the following factors: (1) age (middle aged (12 weeks old) vs. elderly (69 years)) and (2) different animal species (rat vs. human).

Exercise-based CR has shown improvements in quality of life [39], physical function [3], ADL [40], and reduced incidence of clinical events [41] and is becoming increasingly important. Patients with CVD exhibit lower values of various physical functions, such as gait speed [16], 6 MWD [35], and skeletal muscle strength [42], each of which is a predictor of prognosis. Therefore, this study adopted four measures of physical function outcomes: grip strength, QIS, usual gait speed, and 6 MWD. Thus, another strength of this study was its multifaceted examination of the effects of CR.

This study has some limitations. First, this was a single-center observational study and external validity could not be guaranteed. Second, there might have been a selection bias since we included patients in whom the intervention was possible for 5 months in outpatient CR. Additionally, the number of patients in the SGLT2-inhibitors-used group was small; thus, analyses to match patient backgrounds between the two groups, such as propensity matching, were difficult to perform and so was subgroup analyses in patients with sarcopenia or frailty. SGLT2 inhibitors have been shown to improve prognosis, even in frail patients [43], although further studies are required to determine their association with physical function.

## 5. Conclusions

The use of SGLT2 inhibitors in patients with CVD and T2DM undergoing outpatient CR did not impair improvement in physical functions. Further investigation remains warranted to determine the impact on the improvement in physical function in patients with CVD, especially those with sarcopenia and frailty.

## Figures and Tables

**Figure 1 jcm-11-05956-f001:**
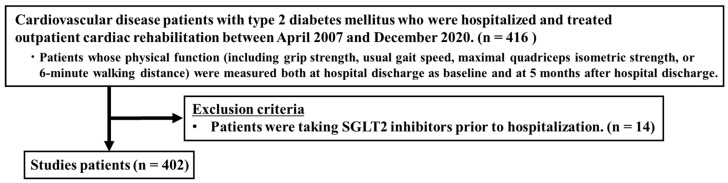
Flow chart in this study. SGLT2, Sodium-Glucose Cotransporter 2.

**Figure 2 jcm-11-05956-f002:**
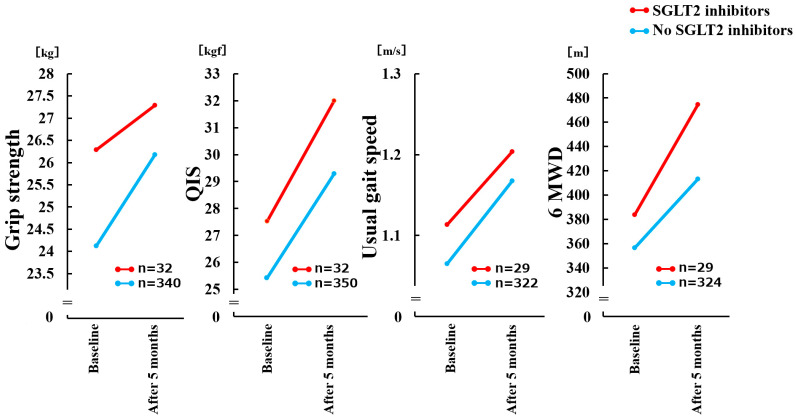
Comparison of Δ physical functions with and without SGLT2 inhibitor use in a mixed-effects model. There was no significant time × group interaction (*p* = 0.178) for change in Grip strength. There was no significant time × group interaction (*p* = 0.575) for change in QIS. There was no significant time × group interaction (*p* = 0.757) for change in Usual gait speed. There was no significant time × group interaction (*p* = 0.151) for change in 6 MWD. QIS, maximal quadriceps isometric strength; 6 MWD, 6-minute walking distance; SGLT2, Sodium-Glucose Cotransporter 2.

**Table 1 jcm-11-05956-t001:** Baseline characteristics of patients using and not using SGLT2 inhibitors.

Variables	Overall	SGLT2 Inhibitors	Non SGLT2 Inhibitors	*p*-Value
	*n* = 402	*n* = 32	*n* = 370	
Age [years]	69 [60–76]	64 [52–68]	69 [61–76]	0.002
Older (≥65), *n* (%)	262 (65.2)	14 (43.8)	248 (67.0)	0.011
Males, *n* (%)	274 (68.2)	21 (65.6)	253 (68.4)	0.843
body weight [kg]	59.2 [50.3–67.7]	60.7 [50.9–69.5]	59.2 [50.1–67.4]	0.525
BMI [kg/m^2^]	22.6 [20.2–25.1]	23.2 [20.1–25.8]	22.6 [20.2–25.0]	0.569
LVEF [%]	54.2 [42.7–64.5]	42.0 [29.0–58.8]	55.0 [44.0–65.0]	0.009
The total number of outpatient CR [times]	4.0 [3.0–8.0]	3.0 [2.0–6.5]	4.5 [3.0–8.0]	0.059
Diagnosis, *n* (%)				
ACS	128 (31.8)	12 (37.5)	116 (31.4)	0.553
HF	123 (30.6)	16 (50.0)	107 (28.9)	0.017
Others	151 (37.6)	4 (12.5)	147 (39.7)	0.002
History of HF hospitalization, *n* (%)	74 (18.4)	6 (18.8)	68 (18.4)	1.000
History of AMI, *n* (%)	70 (17.4)	2 (6.2)	68 (18.4)	0.09
Hb [g/dL]	12.3 [10.6–14.0]	14.2 [12.5–15.4]	12.2 [10.5–13.7]	<0.001
BNP [pg/mL]	177 [62–512]	268 [79–741]	176 [58–476]	0.221
eGFR [mL/min/1.73 m^2^]	57 [39–71]	60 [48–74]	57 [38–71]	0.285
Baseline Physical function				
grip strength [kg]	23.4 [16.1–30.7]	27.8 [19.0–31.0]	22.9 [16.1–30.5]	0.143
QIS [kgf]	23.8 [15.5–33.5]	27.1 [16.9–35.7]	23.6 [15.5–33.2]	0.321
usual gait speed [m/s]	1.10 [0.91–1.26]	1.15 [0.98–1.34]	1.08 [0.90–1.26]	0.334
6 MWD [m]	422 [310–498]	440 [360–500]	418 [308–495]	0.476
Medications, *n* (%)				
ACE Inhibitor	159 (39.6)	18 (56.2)	141 (38.1)	0.058
ARB	182 (45.3)	11 (34.4)	171 (46.2)	0.267
Beta-Blocker	310 (77.1)	28 (87.5)	282 (76.2)	0.189

SGLT2, Sodium-Glucose Cotransporter 2; BMI, body mass index; LVEF, left ventricular ejection fraction; CR, cardiac rehabilitation; ACS, acute coronary syndrome; HF, heart failure; AMI, acute myocardial infarction; Hb, Hemoglobin; BNP, B-type natriuretic peptide; eGFR, estimated glomerular filtration rate; QIS, maximal quadriceps isometric strength; 6 MWD, 6-minute walking distance; ACE, angiotensin converting enzyme; ARB, angiotensin II receptor blocker. Values are median [interquartile range]; *n* (%).

**Table 2 jcm-11-05956-t002:** Association of physical functions with SGLT2 inhibitors use.

Variables	Δ Grip Strength		Δ QIS		Δ Usual Gait Speed		Δ 6 MWD	
	*β*	*p*-Value	*β*	*p*-Value	*β*	*p*-Value	*β*	*p*-Value
SGLT2 inhibitors use	0.148	0.061	0.018	0.830	0.129	0.161	0.020	0.822
Age	−0.416	<0.001	−0.359	<0.001	−0.225	0.023	−0.460	<0.001
Male	−0.409	<0.001	−0.238	0.007	0.062	0.497	−0.042	0.626
BMI	0.168	0.021	0.122	0.131	0.059	0.466	−0.042	0.594
NYHA class ≥ III	−0.055	0.463	−0.077	0.312	−0.057	0.503	−0.043	0.602
LVEF	−0.096	0.231	0.063	0.457	−0.111	0.252	0.035	0.699
HF	0.082	0.271	0.017	0.832	0.156	0.079	0.112	0.186
Hb	0.045	0.604	0.261	0.004	0.084	0.413	0.098	0.317
Year of hospitalization (≥2014)	−0.118	0.106	−0.042	0.587	−0.051	0.548	−0.163	0.048
Baseline each physical function	−0.700	<0.001	−0.472	<0.001	−0.376	0.001	−0.462	<0.001

QIS, maximal quadriceps isometric strength; 6 MWD, 6-minute walking distance; SGLT2, Sodium-Glucose Cotransporter 2; BMI, body mass index; NYHA, New York Heart Association; LVEF, left ventricular ejection fraction; HF, heart failure; Hb, Hemoglobin.

## Data Availability

The datasets used and/or analyzed during the current study are available from the corresponding author on reasonable request.

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
