# Peer review of "Association between SGLT2 Inhibitors and Cardiac Rehabilitation Outcomes in Patients with Cardiovascular Disease and Type 2 Diabetes Mellitus"

_jcm, 2022, doi:10.3390/jcm11195956_

Round 1

Reviewer 1 Report

The authors aimed to investigate the effect of SGLT2 inhibitors on the outcomes of patients with CVD and T2DM undergoing comprehensive outpatient CR. There are some issues here, as follows.

1. The time span of the study population is too long and the variation in the treatment of cardiovascular disease may affect the outcome. The use of medications outside the hospital should also be included in the analysis.

2. The population difference between the SGLT2i(+) and SGLT2i(-) groups was too large; either screening patients from the SGLT2i(-) group to match the inclusion time point of the SGLT2i(+) group or using propensity score matching to reduce the inclusion difference between the two groups.

3. Physical function as an outcome includes multiple indicators (grip strength, QIS, usual gait speed, and 6 MWD), and the number of patients in the SGLT2i (+) group is inherently small; still, the exact number of patients completing each indicator needs to be clarified.

4. Exercise-based CR improves prognosis in patients with CVD+T2DM, how can the study draw this conclusion when it lacks controls? In the context of CR, there was no difference in outcome between groups with or without SGTL2i application. can the effect of the SGLT2i(+) group be interpreted as a combined effect of the negative phase effect of SGLT2i on T2DM patients and the positive effect of SGLT2i on CVD. The effect of SGLT2i alone on CVD patients is missing. The specific type of CVD patients in this group also needs to be demonstrated. If the CVD patients are predominantly HF, wouldn't the conclusion be interpreted as the SGLT2i(+) group improving exercise tolerance in HF patients?

Author Response

1. The time span of the study population is too long and the variation in the treatment of cardiovascular disease may affect the outcome. The use of medications outside the hospital should also be included in the analysis.

Response: Thank you for your comment. Indeed, drugs for the treatment of cardiac disease are evolving; we believe that this part of the process must be taken into consideration. Unfortunately, we are unable to fully track changes in outpatient medications. We added the major medications at discharge, ACE inhibitor, ARB, and Beta-Blocker, to Table 1.

2. The population difference between the SGLT2i (+) and SGLT2i (-) groups was too large; either screening patients from the SGLT2i (-) group to match the inclusion time point of the SGLT2i (+) group or using propensity score matching to reduce the inclusion difference between the two groups.

Response: Thank you for your comment. The propensity score matching was difficult due to the small sample size in this study. We added the following to the revised manuscript in the limitations section.

 Line276-279 (Page7, line25-28): Discussion

Before

In addition, due to the small number of prescriptions, sub-analysis of sarcopenia and frail patients was unfortunately not possible.

After

Additionally, the number of patients in the SGLT2 inhibitors used group was small; thus, analyses to match patient backgrounds between the two groups, such as propensity matching, were difficult to perform, and so was subgroup analyses in patients with sarcopenia or frailty.

3. Physical function as an outcome includes multiple indicators (grip strength, QIS, usual gait speed, and 6 MWD), and the number of patients in the SGLT2i (+) group is inherently small; still, the exact number of patients completing each indicator needs to be clarified.

Response: Thank you for your comment. Indeed, the sample sizes for each indicator vary slightly. The actual number of cases used in the analysis is now shown in Figure 2.

4. (1) Exercise-based CR improves prognosis in patients with CVD+T2DM, how can the study draw this conclusion when it lacks controls?

Response (1): Thank you for your comment. We could not assert that CR contributed to the improvement in physical function in this study. We revised the conclusion as follows:

 Changes:

Line283-286 (Page7, line32-35) : Conclusion

Before

Exercise-based CR improved a variety of physical functions in patients with CVD and T2DM, but the amount of improvement was not different between patients with and without SGLT2 inhibitor use. These findings suggest that the use of SGLT2 inhibitors in patients with CVD and T2DM does not interfere with CR-induced improvement in physical functions.

After

The use of SGLT2 inhibitors in patients with CVD and T2DM undergoing outpatient CR did not impair improvement in physical functions. Further investigation remains warranted to determine the impact on the improvement of physical function in patients with CVD, especially those with sarcopenia and frailty.

(2) In the context of CR, there was no difference in outcome between groups with or without SGTL2i application. can the effect of the SGLT2i (+) group be interpreted as a combined effect of the negative phase effect of SGLT2i on T2DM patients and the positive effect of SGLT2i on CVD. The effect of SGLT2i alone on CVD patients is missing.

Response (2): In our study, among CVD patients with DM, there was no difference in the outcome between the CR participants with and without SGTL2 inhibitor treatment. The mechanism underlying the change in physical function with SGTL2 inhibitor use remains unclear since we could not track skeletal muscle mass or nutritional indices and compare the results with those of the CR-naive group. As for the possible mechanism, we considered the combined effect of the negative effect of SGTL2 inhibitor on T2DM patients and the positive effect of SGTL2 inhibitor on CVD based on previous findings; this aspect would be difficult to address in this study hence further investigations are warranted.

(3) The specific type of CVD patients in this group also needs to be demonstrated. If the CVD patients are predominantly HF, wouldn't the conclusion be interpreted as the SGLT2i (+) group improving exercise tolerance in HF patients ?

Response (3): The diagnosis of CVD was added to Table 1. Indeed, the SGTL2 inhibitor (+) group included significantly more patients with HF, and to condier the effect of SGTL2 inhibitor on improving exercise tolerance in patients with HF, we reanalyzed the multiple regression analysis, adding the presence or absence of heart failure as a contributing factor. The results confirmed the lack of association between SGTL2 inhibitor and improvement in physical function, including 6MWD.

These results have been added to Table 2 of the revised manuscript.

Variables

Δ grip strength

Δ QIS

Δ usual gait speed

Δ 6MWD

β

P-value

β

P-value

β

P-value

β

P-value

SGLT2 inhibitors

0.148

0.061

0.018

0.830

0.129

0.161

0.020

0.822

Age

-0.416

< 0.001

-0.359

< 0.001

-0.225

0.023

-0.460

< 0.001

Male

-0.409

< 0.001

-0.238

0.007

0.062

0.497

-0.042

0.626

BMI

0.168

0.021

0.122

0.131

0.059

0.466

-0.042

0.594

NYHA class ≥ â…¢

-0.055

0.463

-0.077

0.312

-0.057

0.503

-0.043

0.602

LVEF

-0.096

0.231

0.063

0.457

-0.111

0.252

0.035

0.699

Heart failure

0.082

0.271

0.017

0.832

0.156

0.079

0.112

0.186

Hb

0.045

0.604

0.261

0.004

0.084

0.413

0.098

0.317

Year of hospitalization (≥ 2014)

-0.118

0.106

-0.042

0.587

-0.051

0.548

-0.163

0.048

Baseline each physical function

-0.700

< 0.001

-0.472

< 0.001

-0.376

0.001

-0.462

< 0.001

QIS, maximal quadriceps isometric strength; 6MWD, 6-minute walking distance; SGLT2, Sodium-Glucose Cotransporter 2; BMI, body mass index; NYHA, New York Heart Association; LVEF, left ventricular ejection fraction; HF, heart failure; Hb, Hemoglobin

Reviewer 2 Report

I read with interest the article "Association between SGLT2 inhibitors and Cardiac Rehabilitation Outcomes in Patients with Cardiovascular Disease and 3 Type 2 Diabetes Mellitus." Unfortunately, there are numerous methodological shortcomings; 1. A very small number of patients (416 pts.) compared to the analysis period of even 13 years! (2007 – 2020) 2. There is a big difference in the number of patients with and without SGLT2 in therapy 3. Should the reason for the introduction of SGLT2 be explained – unregulated diabetes and/or improvement of CHF symptoms? 4. Describe in detail the analyzed group of CVD patients (MI, PCI, CABG, valvular disease and cardiac surgery, etc.) 5. The EU and the FDA approved the use of the first SGLT2 inhibitors 10 years ago - how have patients been included since 2007?? 6. The follow-up period is too short 7. It would be better if NTproBNP was determined rather than BNP 8. There is a significant difference in LVEF, but there is no significant difference in the level of BNP between the observed groups? How do the authors explain it?

Author Response

1. A very small number of patients (416 pts.) compared to the analysis period of even 13 years! (2007 – 2020)

Response: Thank you for your comment. The main reason for the small number of patients compared to the 13-year analysis period is that although we strongly recommend participation in outpatient rehabilitation at the time of discharge from the hospital, unfortunately, many do not participate due to transfer or distance from the hospital, resulting in a small number of cases where functional assessment at two-time points is possible. This is a limitation in this study as the situation is similar not only in Japan but also abroad. However, few studies investigating the effects of SGTL2 inhibitor on physical function have reported effectiveness in many CVD categories; we believe that this is an important study despite the small sample size.

2. There is a big difference in the number of patients with and without SGLT2i in therapy

Response: Thank you for your comment. During this inclusion period, SGTL2 inhibitor prescriptions were not approved for the treatment of HF; many patients are still using other medications for DM. SGTL2 inhibitor has recently been prescribed to many CVD or CKD patients, and studies are ongoing on this topic, including patients with sarcopenia and frailty complications.

3. Should the reason for the introduction of SGLT2 be explained – unregulated diabetes and/or improvement of CHF symptoms?

Response: Thank you for your question. Although empagliflozin was the first SGTL2 inhibitor to be approved for the treatment of HF in Japan in November of 2020, SGTL2 inhibitor in this study population was prescribed as a an anti-diabetic agent since the patients in this study were included before that date.

4. Describe in detail the analyzed group of CVD patients (MI, PCI, CABG, valvular disease and cardiac surgery, etc.)

Response: Thank you for your suggestion. Please see the response to Reviewer 1, Comment 4.

5. The EU and the FDA approved the use of the first SGLT2 inhibitors 10 years ago - how have patients been included since 2007 ?

Response: As you pointed out, the use of SGTL2 inhibitor in diabetic patients in Japan has been approved since 2014, hence patients included in the SGTL2 inhibitor group were those included after that year. To consider the effect of this difference in the inclusion period on the results of this study, the year of inclusion ( ≥ 2014; yes or no) was added as one of the independent variables in the multiple regression analyses. The results of this study did not significantly change, and we found no association between SGTL2 inhibitor use and changes in physical function. We have reflected these changes in the statistical analysis section and the results of the multiple regression analysis (Table 2) in the revised manuscript.

6. The follow-up period is too short

Response: Thank you for your comment. Although previous studies on the effects of exercise therapy on cardiac disease with physical function as the outcome required long follow-up periods of years, other studies with shorter follow-up periods of 3 to 6 months have been reported [1]. In this case, 5 months is reasonable since the decrease in skeletal muscle mass due to SGTL2 inhibitor is a short-term change, and are considered important for concerns about physical function decline [2-4].

7. It would be better if NTproBNP was determined rather than BNP

Response: Thank you for your suggestion, we have been measuring NTproBNP at our institution since the introduction of ARNI as a heart failure therapy, but was not measured in most cases during the inclusion period of this study. Therefore, it is difficult to show in this study.

8. There is a significant difference in LVEF, but there is no significant difference in the level of BNP between the observed groups? How do the authors explain it ?

Response: Thank you for your question. Although the exact reasons are not clear, we believe that such a gap could occur because this study included not only patients with HF but also other patients with CVD, as shown in Table 1 in the revised manuscript.

References

  1. Nelson, M.B.; Gilbert, O.N.; Duncan, P.W.; Kitzman, D.W.; Reeves, G.R.; Whellan, D.J.; Mentz, R.J.; Chen, H.; Hewston, L.A.; Taylor, K.M.; et al. Intervention Adherence in REHAB-HF: Predictors and Relationship With Physical Function, Quality of Life, and Clinical Events. J Am Heart Assoc 2022, 11, e024246, doi:10.1161/jaha.121.024246.
  2. Koike, Y.; Shirabe, S.I.; Maeda, H.; Yoshimoto, A.; Arai, K.; Kumakura, A.; Hirao, K.; Terauchi, Y. Effect of canagliflozin on the overall clinical state including insulin resistance in Japanese patients with type 2 diabetes mellitus. Diabetes Res Clin Pract 2019, 149, 140-146, doi:10.1016/j.diabres.2019.01.029.
  3. Sano, M.; Meguro, S.; Kawai, T.; Suzuki, Y. Increased grip strength with sodium-glucose cotransporter 2. J Diabetes 2016, 8, 736-737, doi:10.1111/1753-0407.12402.
  4. Kitzman, D.W.; Whellan, D.J.; Duncan, P.; Pastva, A.M.; Mentz, R.J.; Reeves, G.R.; Nelson, M.B.; Chen, H.; Upadhya, B.; Reed, S.D.; et al. Physical Rehabilitation for Older Patients Hospitalized for Heart Failure. N Engl J Med 2021, 385, 203-216, doi:10.1056/NEJMoa2026141.

Round 2

Reviewer 1 Report

I consider that the requirements have been met